# Developing Endogenous Autophagy Reporters in *Caenorhabditis elegans* to Monitor Basal and Starvation-Induced Autophagy

**DOI:** 10.3390/ijms262010178

**Published:** 2025-10-20

**Authors:** Kincső Bördén, Tibor Vellai, Tímea Sigmond

**Affiliations:** 1Department of Genetics, Eötvös Loránd University (ELTE), Pázmány Péter sétány 1/C, 1117 Budapest, Hungary; 2Hungarian Research Network-ELTE, Genetics Research Group, 1117 Budapest, Hungary

**Keywords:** ATG-5, autophagy, *C. elegans*, endogenous reporters, LGG-1, TOR, starvation

## Abstract

Autophagy (cellular self-eating) is a tightly regulated catabolic process of eukaryotic cells during which parts of the cytoplasm are sequestered and subsequently delivered into lysosomes for degradation by acidic hydrolases. This process is central to maintaining cellular homeostasis, the removal of aged or damaged organelles, and the elimination of intracellular pathogens. The nematode *Caenorhabditis elegans* has proven to be a powerful genetic model for investigating the regulation and mechanism of autophagy. To date, the fluorescent autophagy reporters developed in this organism have predominantly relied on multi-copy, randomly integrated transgenes. As a result, the interpretation of autophagy dynamics in these models has required considerable caution due to possible overexpression artifacts and positional effects. In addition, starvation-induced autophagy has not been characterized in detail using these reporters. Here, we describe the development of two endogenous autophagy reporters, *gfp::mCherry::lgg-1/atg-8* and *gfp::atg-5*, both inserted precisely into their endogenous genomic loci. We demonstrate that these single-copy reporters reliably track distinct stages of the autophagic process. Using these tools, we reveal that (i) the transition from the earliest phagophore to the mature autolysosome is an exceptionally rapid event because the vast majority of the detected fluorescent signals are autolysosome-specific, (ii) starvation triggers autophagy only after a measurable lag phase rather than immediately, and (iii) the regulation of starvation-induced autophagy depends on the actual life stage, and prevents excessive flux that could otherwise compromise cellular survival. We anticipate that these newly developed reporter strains will provide refined opportunities to further dissect the physiological and pathological roles of autophagy in vivo.

## 1. Introduction

Autophagy is a fundamental catabolic process in eukaryotic cells, whereby cytoplasmic material destined for degradation is sequestered and delivered to lysosomes. Depending on the type of autophagy, this occurs through distinct mechanisms. Once inside the lysosome, macromolecules are broken down by acidic hydrolases, and the resulting components are recycled to fuel anabolic pathways and energy production [1,2,3]. Autophagy is therefore indispensable for maintaining cellular homeostasis [4], removing dysfunctional organelles [5], enabling appropriate stress responses [6], eliminating intracellular pathogens [7,8,9], and modulating the pace of aging [10]. Impairments in autophagy have been linked to a wide array of degenerative conditions, including cancer, various neurodegenerative diseases, diabetes, tissue atrophy, and fibrosis [11,12,13,14,15]. While certain autophagy pathways are organelle-selective (e.g., mitophagy or pexophagy) [16], autophagy is mechanistically distinguished from the ubiquitin–proteasome system (UPS). The UPS specifically recognizes and degrades ubiquitin-tagged proteins, whereas autophagy non-specifically sequesters bulk regions of the cytoplasm for lysosomal degradation [17].

Three principal forms of autophagy have been described: microautophagy, chaperone-mediated autophagy, and macroautophagy. Among these, macroautophagy (hereafter referred to simply as “autophagy”) is the most prominent, characterized by the encapsulation of cargo within a double-membraned vesicle known as the autophagosome. Subsequent fusion of the autophagosome with a lysosome yields an autolysosome, where enzymatic digestion of the cargo occurs. The autophagic process progresses through sequential stages—initiation, nucleation, expansion, lysosomal fusion, degradation, and recycling.

The master regulator of autophagy is the target of rapamycin (TOR) protein kinase. Under nutrient-rich conditions, TOR suppresses autophagy by inhibiting the initiation complex, whereas during starvation, this inhibition is released. TOR exerts its regulatory effects through two main mechanisms: (i) hyperphosphorylation of key initiation complex components, thereby altering protein–protein interactions and preventing autophagosome nucleation, and (ii) modulation of downstream signaling cascades that govern the transcriptional and translational regulation of autophagy-related proteins [3,18,19].

The nematode *Caenorhabditis elegans* represents an excellent experimental model for studying autophagy. Its small size, short life cycle (reaching reproductive maturity in ~3 days at 22 °C), and ease of laboratory maintenance make it particularly tractable. The transparency of its cuticle and the invariant cell lineage of development allow cell-specific tracking of fluorescently tagged transgenes [20]. Furthermore, the predominance of hermaphrodites ensures the stable propagation of isogenic strains, while genetic crosses allow the generation of complex mutant backgrounds.

In *C. elegans*, the autophagic response to starvation is mediated primarily through the TOR ortholog LET-363, which integrates signals from the insulin/IGF-1-like signaling (IIS) pathway. In nutrient-rich conditions, LET-363 represses autophagy, while under starvation, this repression is alleviated. Components of the IIS pathway, including the IGF-1 receptor homolog DAF-2 and the FoxO transcription factor homolog DAF-16, serve as critical modulators of this nutrient-sensitive regulation [21,22].

Fluorescent protein reporters remain one of the most widely used tools for monitoring autophagy. Typically, autophagy-related proteins are fused to green fluorescent protein (GFP) or to red fluorescent proteins such as mCherry, enabling visualization of autophagic structures [23,24]. Among these, the ATG8 homolog LGG-1 has been extensively used, as it associates with both nascent and mature autophagic membranes, thereby serving as a robust marker of autophagosome and autolysosome dynamics [25]. However, current reporter systems are subject to several caveats, including multi-copy insertion at random genomic sites, variable expression, and pH-sensitivity of GFP that limits detection of late-stage autophagic compartments [26,27,28,29,30,31].

Meléndez et al. were the first to generate a *C. elegans* strain expressing GFP::LGG-1 (DA2123: *adIs2122* [*lgg-1p::GFP::lgg-1* + *rol-6(su1006)*]) [25]. The number of GFP::LGG-1-positive structures changes dynamically throughout all developmental stages of *C. elegans* under normal conditions. However, this strain also displays strong background fluorescence, which complicates the detection of specific structures. Under conditions influencing autophagic activity, such as starvation, the number of GFP::LGG-1-positive foci slightly increases [26], though often only negligibly and with high variability. While the number of GFP::LGG-1-labeled foci can indicate an increase in autophagic activity, the accumulation of autophagic structures may also result from a blockage in the process [27,28]. Because GFP is highly sensitive to acidic pH [29], this reporter strain labels only pre-autophagosomal structures and autophagosomes, providing no information about autolysosomes (Figure 1) [30]. Autophagosome formation itself is a rapid process, and thus measurable autophagic activity quickly subsides after induction [27]. Finally, the transgene is integrated as a multicopy array at a single genomic locus. Taken together, tracking the GFP::LGG-1 signal alone does not provide sufficient evidence regarding the direction or nature of changes in autophagic activity.

Chang et al. inserted *mCherry* gene into the LGG-1 reporter construct (MAH215: *sqIs11* [*lgg-1p::mCherry::GFP::lgg-1* + *rol-6*]) [27]. Dual labeling with GFP and mCherry enables not only the visualization of autophagosomes and autolysosomes but also their distinction. Specifically, autophagosomes appear yellow due to the combined GFP and mCherry signals, whereas autolysosomes appear only red, since GFP is unstable under the acidic pH of lysosomes and is degraded following autophagosome–lysosome fusion, resulting in the loss of the green signal from these structures (Figure 1) [27,29].

To overcome these limitations, we set out to develop a precise, endogenous single-copy autophagy reporter system in *C. elegans*. Specifically, we generated a GFP::mCherry::LGG-1 reporter line using CRISPR–Cas9 genome editing, and additionally designed a GFP::ATG-5 reporter, given the essential role of ATG-5 in autophagosome formation and subsequent lysosomal fusion [18,28]. We validated these reporters by monitoring starvation-induced autophagy at multiple developmental stages, and further tested their responsiveness to mutations and RNA interference, as well as pharmacological inhibition using bafilomycin A1.

## 2. Results

### 2.1. The Endogenous GFP::mCherry::LGG-1/ATG-8 Reporter Is Expressed Ubiquitously in *C. elegans* Tissues from Early Embryonic Stages Onward

In this study, we used GFP::mCherry::LGG-1 and GFP::ATG-5 strains generated with the CRISPR/Cas9 system (exact genotypes are listed in the Materials and Methods section, the sequences in Appendix A). The strains were created by microinjection, performed by Suny Biotech. To eliminate background mutations, the strains were backcrossed twice with the wild-type genetic background (isogenization).

The endogenous GFP::mCherry::LGG-1 reporter construct was designed so that the transgene was integrated immediately after the start codon of the endogenous *lgg-1* gene. The construct consists of GFP and mCherry sequences linked by a short, glycine-rich linker region (Figure 2A). This linker region helps ensure that the LGG-1 protein can function properly even in the presence of the attached fluorescent proteins [32]. In the reporter strain, Western blot analysis confirmed the presence of the GFP::mCherry::LGG-1 fusion protein, which was not detectable in wild-type animals (Appendix A). The fusion protein is characterized by diffuse GFP distribution and punctate mCherry-positive structures (Figure 2B,B’). In adult animals, the protein is expressed in all tissue types, though the fluorescent signal is relatively weak due to its single-copy nature. In the intestine, strong non-specific autofluorescence was observed, which prevented meaningful analysis of autophagic structures in this region. Therefore, our experiments were largely restricted to the head region, where no other fluorescent signals interfered with detection (Figure 2B’,B”).

Expression analyses revealed that the protein is expressed in accordance with the endogenous *lgg-1* gene, beginning at the 300-cell stage of embryonic development (Figure 2C). In comma-stage embryos, apoptotic cells co-localized with mCherry foci (Figure 2C’). In adult animals, the protein is expressed in all cell types, including the somatic distal tip cells (DTC) of the gonad as well as in germ cells (Figure 2D,D’). Together, these results confirm that the GFP::mCherry::LGG-1 reporter reflects the endogenous expression and localization of LGG-1, making it a reliable tool for monitoring autophagy in *C. elegans*.

### 2.2. The Endogenous GFP::mCherry::LGG-1 Reporter Responds to Autophagy-Modulating Stimuli

In the GFP::mCherry::LGG-1 reporter strain, we performed multiple experiments to observe changes in autophagic activity under different inducing or inhibiting conditions. During our analyses, we measured the percentage coverage of GFP and mCherry foci in the head region of animals, where intestinal autofluorescence did not interfere with detection. As described previously, GFP and mCherry double-positive structures represent autophagosomes, whereas mCherry-only puncta indicate the presence of autolysosomes (Figure 1). An increase in the number of autophagosomes may reflect elevated autophagic activity; however, such an increase can also result from a blockage at downstream stages of the process, for example, failure of autophagosome–lysosome fusion [27,33]. Similarly, autolysosome accumulation may also indicate a downstream defect [34]; nonetheless, the generally accepted view is that the abundance of autolysosomes correlates with autophagic capacity [28].

In transgenic nematodes with an otherwise wild-type genetic background, we observed only red-labeled structures, while GFP signal was hardly detectable at all (Figure 2B–B”). This phenomenon may be explained by the fact that the autophagosome-to-autolysosome transition is an extremely rapid process, and therefore only a very small number of autophagosomes can be observed at any given time [27].

In adult animals, we examined GFP and mCherry coverage in wild-type versus *epg-5(tm3425)* mutant genetic backgrounds. The *epg-5* gene is required for autophagosome–lysosome fusion. In its functional absence, fusion occurs only to a limited extent, leading to autophagosome accumulation. Tien et al. demonstrated that not only the loss of *epg-5* but also its overexpression significantly disrupts normal autophagic activity, indicating that *epg-5* activity determines the overall rate of autophagy [35,36]. We found that in the *epg-5(tm3425)* loss-of-function mutant background, the GFP-covered area was comparable to that of wild type, while mCherry coverage was significantly reduced, confirming a defect in the fusion step (Figure 3B–B”). *hlh-30* encodes the *C. elegans* TFEB transcription factor, the master regulator of autophagy genes, which induces many autophagy-related genes, particularly under stress conditions [37,38]. In *hlh-30(-)* loss-of-function mutants, both GFP and mCherry signals showed low expression, reflecting reduced autophagic activity (Figure 3C–C”). Thus, in the absence of the *epg-5* and *hlh-30* gene functions required for proper autophagy, we observed reduced fluorescence labeling indicative of decreased autophagic activity compared to the wild-type genetic background. Taken together, these results highlight that the GFP::mCherry::LGG-1 reporter is particularly well suited to detect disruptions in autophagosome–lysosome fusion and reduced autophagic flux in mutant backgrounds.

### 2.3. Silencing of *Let-363/TOR* Increases the Number of Autolysosomes in the Endogenous GFP::mCherry::LGG-1 Reporter Strain

We also tested autophagic activity, as measured by the transgene, in a *let-363/TOR(-)* deficient genetic background. In this experiment, the autophagy-inhibiting gene *let-363/TOR* was silenced using RNAi feeding. As a key regulator of autophagy, LET-363/TOR provides an excellent model for studying regulatory interactions [21,38]. *let-363/TOR* RNAi-treated animals were compared with control animals fed with HT115 bacteria at the L1, L2, and L3 larval stages. RNA interference had no effect on the F1 generation, which had been exposed to RNAi bacteria from the onset of hatching; however, continued *let-363* RNAi treatment in the F2 generation resulted in differences compared to *C. elegans* larvae fed with HT115 bacteria (Figure 3D–D”). In L1 and L2 larvae, we observed a significant increase in the number of autophagic structures labeled by the mCherry reporter in *let-363/TOR(RNAi)* animals compared to controls. By contrast, in the L3 larval stage, no difference was detected, although development was arrested at this stage (L3 “arrest”). The phenomenon of L3 arrest following autophagy inhibition was previously described by Long et al., though not specifically in the F2 generation [39].

As described earlier, the number of mCherry-positive autolysosomes is generally considered to correlate directly with autophagic activity [40]. Thus, the differences observed between control and RNAi-treated animals strongly support the conclusion that the product of the *let-363/TOR* gene inhibits autophagy, and that in its absence, autophagic activity increases [3,18,19,38].

### 2.4. Starvation Increases the Number of Autolysosomes in L1 Larvae and Adult Animals

Starvation stress is most commonly studied in L1 arrest and young adult animals [41]. The L1 arrest state is triggered by lack of food immediately after hatching. In this state, development and cell division, and consequently aging, are halted. Upon refeeding, development resumes after a short recovery period [42]. Survival during L1 arrest depends on autophagic activity [43,44]. An advantage of measuring autophagy in the L1 stage compared to adults is that the assay can be easily standardized, allows analysis of large numbers of individuals, and avoids complications caused by progeny production [41,44]. During L1 arrest, we monitored changes in autophagic activity using the endogenous GFP::mCherry::LGG-1 reporter in wild-type and *epg-5(tm3425)* loss-of-function backgrounds. The effect of starvation was assessed in the head region of the larvae, with fed L1s serving as controls. After 24 h of starvation, no punctate GFP signal indicative of autophagosomes was detectable in the head region of L1 animals (Figure 4A,A’). By contrast, the number of lysosomes on the first day of starvation increased threefold, reflecting preceding autophagic activity. With further starvation, we observed a decrease in the mCherry-positive area in the surviving larvae.

In the *epg-5(tm3425)* loss-of-function mutant background, the high GFP signal observed in fed controls indicates autophagosome accumulation. During starvation, this value decreased significantly (Figure 4B,B’), but still consistently reflected the lack of fusion in all animals. The continuous decline in mCherry signal further confirmed the slow elimination of autophagic structures in the *epg-5(-)* mutant background.

We also monitored the effect of starvation in young adult hermaphrodites using the reporter construct. In contrast to the L1 stage, autophagic activity persisted longer in starved adults. During the first 1–5 h, no changes were observed. Previous studies have likewise reported that autophagic activity in adults does not change immediately, requiring at least ~10 h [35], and in some cases up to two days, before significant changes become apparent [44]. After 24 h, GFP-positive puncta indicative of autophagosomes were still detectable, but their number was considerably lower compared to controls. By this time, the number of mCherry-positive areas marking autolysosomes was significantly higher than in controls (Figure 4C,C’), reflecting a preceding autophagic burst. The slow decline of the process was further supported by the continued increase in mCherry-positive signal over the next two days, followed by the elimination of autophagic structures in subsequent days. Consistent with previous reports, we also found that autophagic activity remained unchanged during the early hours of starvation, as the endogenous reporter showed no difference within the first 1–3 h after food withdrawal [44].

Regarding the effect of starvation, we asked whether the process is reversible—that is, whether the proportion of autophagic structures induced by starvation decreases again once the worms are returned to nutrient-rich conditions. We found that this process is indeed reversible. L1 larvae were subjected to starvation for five days and then refed. Within just one hour, the proportion of autophagic structures returned to pre-starvation levels and remained so with continued feeding (Figure 5A–A”).

By performing starvation experiments, we demonstrated that as little as one day of food deprivation significantly increased the proportion of mCherry-positive structures in both L1 larvae and adult wild-type animals, indicating elevated autophagic activity. In contrast, this increase was absent in the *epg-5(tm3425)* mutant background, supporting the role of the *epg-5* gene in autophagosome–lysosome fusion. Furthermore, our experiments with wild-type L1 stage larvae revealed that the starvation-induced process is reversible upon refeeding. Overall, starvation induces a transient autophagic burst that primarily mobilizes the material stored within autolysosomes, while subsequent autophagic activity is temporarily restricted as a protective mechanism, before rapidly resuming recycling upon refeeding.

### 2.5. Effect of the Autophagy Inhibitor Bafilomycin A1 on the Expression of the Endogenous GFP::mCherry::LGG-1 Reporter

Another method suitable for modulating the autophagic process is the application of bafilomycin A1. This compound is a vacuolar H^+^-ATPase inhibitor that prevents protons (H^+^) from entering vacuoles or lysosomes, leading to their accumulation in the cytosol. In recent years, bafilomycin A1 has been increasingly used in autophagy research, as it blocks autophagosome–lysosome fusion and thereby lysosomal degradation in human cell lines [45,46]. To test the effect of bafilomycin A1 in the endogenous reporter strain, we examined how the autophagic response to starvation changed in arrested L1 larvae. In contrast to the untreated control group, where starvation increased the number of mCherry-labeled autophagic structures, bafilomycin A1 treatment abolished this increase in the tested worms (Figure 5B–B”). As a result of the inhibition of autophagosome–lysosome fusion, a moderately elevated number of yellow-colored autophagic structures, positive for both GFP and mCherry, were detected, indicating autophagosome accumulation. It is worth noting that a significant proportion of larvae died during the experiment as early as the first day of bafilomycin A1 treatment. Thus, while starvation alone increased the number of mCherry-positive autolysosomes in L1 larvae, starvation combined with bafilomycin A1 treatment prevented this increase due to the inhibitory effect of bafilomycin A1 on autophagosome–autolysosome fusion. During the experiment, we observed not only a difference in the number of autophagic structures but also the lethality of many bafilomycin A1-treated L1 larvae. While we could observe almost only living L1 larvae among the hundreds of animals that had not been treated with bafilomycin A1, the vast majority of treated larvae died on the first day after the treatment began.

### 2.6. Analysis of Autophagy Using the Endogenous GFP::ATG-5 Reporter

The ATG-5 protein plays an important role during autophagy. As a component of the ATG5–ATG12–ATG16 complex, it contributes to isolation membrane expansion and autophagosome maturation. In addition, it is also involved in autophagosome–lysosome fusion [18,47]. When designing the reporter construct, it was important to place the *gfp* gene upstream of *atg-5*, since it is not known which of the three ATG-5 isoforms is functionally active (Figure 6A). Similar to the other reporter strain we designed, we performed Western blot analysis, which detected the presence of GFP::ATG-5 in reporter-carrying animals but not in wild-type animals (Figure 6B). In experiments using the GFP::ATG-5 strain, we initially examined the head region, as in the GFP::mCherry::LGG-1 strain. However, no punctate structures were detectable, and only a small number were observed in embryos (Figure 6C). We were able to detect only a decrease in GFP intensity during starvation of L1 larvae and adult hermaphrodites (Figure 7A–B’).

Although we were unable to analyze the number of GFP-positive structures in the head region, we performed an experiment in which we compared their numbers in the first intestinal cell of wild-type and *glo-4(ok623)* mutant backgrounds. Based on published results, it is assumed that the *glo-4* gene encodes the guanine exchange factor (GEF) of the GLO-1 Rab GTPase, which is responsible for the formation of intestinal granules in *C. elegans* [48,49]. In the absence of *glo-4* function, these granules disappear from the intestine, thereby facilitating the analysis of autophagic structures. In our experiment, GFP-labeled puncta were observed in the intestine of fed adult animals, but these disappeared upon starvation (Figure 7C,C’). As previously described, GFP::ATG-5 labels only pre-autophagosomal structures and autophagosomes [30]. Therefore, the disappearance of GFP-positive puncta likely indicates the loss of autophagosomes. A reduction—or in this case, complete disappearance—of autophagosomes can serve as a good indicator of increased autophagic activity, reflecting more rapid and efficient autophagosome–lysosome fusion. Under normal conditions, this is already a very fast process, and starvation-induced induction can further accelerate it [27,28].

In *glo-4(ok623); GFP::ATG-5* animals, a few GFP-positive puncta were observed in the first intestinal cells of control, fed young adults, but these disappeared completely upon starvation, which may indicate increased autophagic activity. In sum, the disappearance of GFP::ATG-5 puncta in the intestine upon starvation reflects a rapid autophagosome–lysosome fusion and increased autophagic activity.

## 3. Discussion

Autophagy is a crucial mechanism for proper cellular function, and as such, the study of this process has received considerable attention in biological and medical research. It plays roles in maintaining cellular homeostasis [3,5], mediating stress responses [6], regulating the aging process [10], and eliminating various intracellular pathogens [7]. One of the main regulators of autophagy is the TOR protein, which senses nutrient availability and modulates autophagic activity accordingly [3,18,19]. Autophagy is a multistep process regulated primarily by ATG proteins [18]. During the process, a portion of the cytoplasm is sequestered by a double-membrane autophagosome, and after fusion with the lysosome, the cellular components are degraded in the autolysosome and subsequently recycled back into the cytoplasm [3,18].

Due to its importance, autophagy has been studied from many different perspectives by numerous research groups. In *C. elegans*, one of the most popular topics in autophagy research is aging and lifespan, with many publications devoted to this area. By contrast, the relationship between autophagy and starvation in *C. elegans* remains less well explored, although it has been extensively studied in the fruit fly *Drosophila melanogaster* [50,51,52] and has also been investigated in mammals [53]. Even in *C. elegans* studies that mention starvation, its effect on autophagy is typically not the primary research focus [30,44,54]. For a long time, research relied on tagging a single autophagy-related protein with a fluorophore, which alone was insufficient to provide an accurate picture of changes in autophagic activity. In recent years, however, the use of fluorescent reporter strains has become increasingly widespread. In 2017, Chang et al. published a construct tagged with not one but two fluorophores [27]. This advancement made it possible not only to monitor changes in the number of autophagosomes but also autolysosomes, and to distinguish between the two structures. Their publication reported that the number of autophagosomes increases with age in the intestine, muscles, and pharynx, and to a lesser extent in neurons, while the number of autolysosomes increases until approximately day 3 of adulthood, after which it stagnates in intestinal and pharyngeal cells and decreases in muscles and neurons [27]. These observations correlate with the age-associated decline in autophagic activity [55].

In the present study, we investigated autophagy in *C. elegans*, focusing primarily on its response to starvation, using two endogenous fluorescent reporters that we designed. Our custom reporter constructs allowed us to overcome problems associated with previously published strains. Such issues included the strong background signal of GFP::LGG-1 and its inability to label autolysosomes [30], or in the case of the MAH215 strain (GFP::mCherry::LGG-1) [27], the consequences of excessive expression due to the multicopy nature of the transgene. Our improved reporter system consisted of a GFP::mCherry::LGG-1 strain generated with the CRISPR/Cas9 system [56] and a GFP::ATG-5 strain created using the same technique. Using these strains, we performed starvation and RNA interference experiments, as well as tested the effect of the autophagy inhibitor bafilomycin A1.

Our expression analyses revealed that the LGG-1 protein is expressed from the 300-cell stage of embryonic development (Figure 2C). In adults, expression was observed in all cell types. In the absence of HLH-30, a protein required for starvation-induced activation of autophagy, GFP and mCherry expression were low (Figure 3C–C”), confirming the absence or very low level of autophagic activity. According to our findings, this process subsides during the first 24 h of L1 arrest, consistent with the results of Kang et al. (2007) [44]. The further contribution of autophagic activity to survival is likely ensured by the recycling of autolysosomal contents, as suggested by the decline in autolysosome area observed during prolonged starvation. The decrease in detectable area may also be partly explained by the slower quenching of mCherry; however, this too would occur as a consequence of continuous lysosomal degradation. Strict suppression of autophagic activity may be essential for larvae arrested in early developmental stages. Since *C. elegans* cell lineages follow invariant developmental programs, cell death caused by excessive autophagy would lead to irreparable tissue damage after refeeding. Importantly, no such abnormally developed animals were observed in our experiments. In contrast, starvation in adults did not result in such strict suppression. Even after the first 24 h, low levels of autophagic activity could still be detected during prolonged starvation. This suggests a similar mode of regulation to that observed in the *Drosophila* salivary gland and mammalian muscle cells, where a rapid autophagic response to short-term starvation is replaced by adaptive downregulation during prolonged starvation to protect tissues [57,58].

We also found that not only does the number of autolysosomes increase upon starvation, but this process can also proceed in the opposite direction: in starved larvae carrying the endogenous GFP::mCherry::LGG-1 reporter, the proportion of autolysosomes decreased once the worms were refed (Figure 5A–A”). We observed that silencing the *let-363/TOR* gene in the F2 generation led to a significant increase in the number of autolysosomes in both L1 and L2 larvae (Figure 3D–D”). A similar outcome was obtained in our experiment with the autophagosome–lysosome fusion inhibitor bafilomycin A1: in starved worms treated with bafilomycin A1 [45,46], the proportion of mCherry-positive structures was comparable to that of well-fed worms (Figure 5B–B”). Since changes in the number of autolysosomes are likely to directly correlate with autophagic activity [28], our observations support the broad applicability of this strain for autophagy research. In our experiments with the second newly generated endogenous reporter strain, we found that GFP::ATG-5-positive foci in the intestine disappeared upon starvation (Figure 7C,C’). However, because changes in autophagosome number alone do not unequivocally indicate whether autophagic activity is increasing or decreasing [27,28], this information by itself is not sufficient to conclude that the disappearance of GFP-labeled autophagosomes represents increased autophagy. Taken together with our results using the endogenous GFP::mCherry::LGG-1 reporter, however, a clearer picture emerges: the disappearance of autophagosomes combined with the increase in autolysosomes strongly suggests that starvation induces an intensification of autophagic activity.

Taken together, while further experiments will be necessary in the future to unequivocally validate the applicability of the endogenous GFP::ATG-5 reporter construct, the endogenous GFP::mCherry::LGG-1 reporter construct has proven to be a suitable candidate for studying the process of autophagy and its activity changes in *C. elegans*.

## 4. Materials and Methods

### 4.1. Strains and Maintenance

*C. elegans* strains were maintained following standard protocols as described in WormBook (www.wormbook.org), on nematode growth medium (NGM) plates seeded with *Escherichia coli OP50* at 20 °C. In addition to the wild-type strain, the following reporter lines were generated, using CRISPR–Cas9 genome editing (provided by Suny Biotech): PHX2459 *lgg-1(syb2459) [lgg-1p::gfp::mCherry::lgg-1]* and *PHX2476 atg-5(syb2476) [atg-5p::gfp::atg-5]* (Appendix A). To create genetic combinations, the following crosses were performed: TTV823 *atg-5(syb2476) [atg-5p::gfp::atg-5]*; *glo-4(ok623) V*—derived from crossing PHX2476 with RB811 *glo-4(ok623) V*. TTV801 *lgg-1(syb2459) [lgg-1p::gfp::mCherry::lgg-1]*; *epg-5(tm3425) II*—derived from crossing PHX2459 with FX3425 *epg-5(tm3425) II*. TTV923 *lgg-1(syb2459) [lgg-1p::gfp::mCherry::lgg-1]*; and *hlh-30(tm1978) IV*—derived from crossing PHX2459 with JIN1375 *hlh-30(tm1978) IV*. For RNA interference (RNAi) experiments, *E. coli* HT115 (DE3) was used as the RNAi feeding strain, and *E. coli* DH5α was used for plasmid propagation.

### 4.2. Western Blot Analysis

For protein isolation, 50–100 adult worms were collected in 20 µL of M9 buffer. Cell lysis was performed in the presence of Western blot loading buffer by three freeze–thaw cycles followed by sonication. Samples were then incubated at 95 °C for 5 min. Lysates were stored at −80 °C until further analysis by SDS-PAGE. Western blot analysis was performed using an SDS-containing gradient acrylamide gel, a nitrocellulose blotting membrane, and TBST (Tris-Buffered Saline with Tween 20) as washing solution. A 3% solution of milk powder in TBST served as the blocking reagent. Protein labeling was carried out overnight at 4 °C with the following primary antibodies: anti-GFP (anti-GFP rabbit, 1:2000, a gift from Gábor Juhász, Eötvös Loránd University, Budapest, Hungary), anti-mCherry (anti-mCherry rat, 1:1000, a gift from Gábor Juhász, Eötvös Loránd University, Budapest, Hungary), and anti-α-Tub84B (mouse, 1:2500, Sigma T6199, St. Louis, MO, USA). Secondary antibodies were applied at room temperature for one hour, which included: anti-rabbit IgG alkaline phosphatase (1:1000, Sigma, A3687), anti-mouse IgG alkaline phosphatase (1:1000, Sigma, A8438), and anti-rat IgG alkaline phosphatase (1:1000, Sigma, A8438). Primary and secondary antibodies were removed by three 10-min washes in TBST. After washes, membranes were incubated in AP buffer. For signal visualization, NBT/BCIP (Sigma, 72091) was used, prepared in AP (alkaline phosphatase) buffer in 1:50 dilution.

### 4.3. Starvation Assay

For L1 larval starvation, the first step was to obtain a large number of gravid hermaphrodites. Animals were then washed off NGM plates with M9 buffer (recipe: WormBook, Maintenance of *C. elegans*; Protocol 6) and pelleted by centrifugation at 2000 rpm for 2 min. After removal of the supernatant, 5 mL of bleach solution (recipe: WormBook, Maintenance of *C. elegans*; Protocol 4) was added. Incubated for 8–10 min with occasional vortexing until the adult animals were completely dissolved, leaving behind their eggs. Immediately, 5 mL of M9 was added to stop the reaction. After repeating the centrifugation under the same conditions, the supernatant was removed, and the pellet was washed three times with 10 mL M9 buffer, using the same centrifugation settings. Following the final wash, the eggs were resuspended in 2 mL M9 buffer and incubated at 20 °C for the duration of the experiment. For adult starvation, hermaphrodites at the L4 larval stage were transferred to NGM agar plates lacking OP50 bacterial lawns and containing 25 µg/mL fluorodeoxyuridine (FUDR) to prevent egg-laying and avoid mixing of generations. The animals were maintained at 20 °C throughout the starvation period.

### 4.4. RNA Interference

RNA interference (RNAi) was performed using the feeding method, i.e., animals consumed bacteria containing double-stranded RNA (dsRNA) specific to the target gene intended for silencing [59]. The T444T plasmid carrying the *let-363/TOR* RNAi sequence [60] was first transformed into *E. coli* DH5α and subsequently into the HT115 strain to achieve more stable transformation. In the T444T plasmid, each end of the multicloning sites is flanked not only by T7 promoters but also by T7 terminator sequences, ensuring that only the intended sequence is transcribed, thereby increasing efficiency and specificity [60]. As a control, HT115 bacteria carrying the empty construct were used. For *let-363* RNAi and control experiments, synchronized hermaphrodites were allowed to lay eggs on NGM plates seeded with the appropriate bacterial lawns. After 1.5–2 h, egg-laying adults were removed. Once the F1 generation reached adulthood, they were again placed on fresh RNAi and control plates for egg-laying, and experiments were performed on the resulting F2 progeny.

### 4.5. Bafilomycin A1 Treatment

Starved L1 stage larvae were prepared as described above (starvation assay). Shortly, embryos were collected and dissolved in 100 µL M9 solution. Finally, 0.93 µL Bafilomycin A1 (a specific inhibitor of vacuolar H+-ATPase, which inhibits autolysosome formation) was added to achieve a final concentration of 30 nM. Embryos were allowed to develop into L1 larvae.

### 4.6. Fluorescent Microscopy

Fluorescent images were acquired using a Zeiss Axioimager Z1 upright microscope (Carl Zeiss Industrielle Messtechnik GmbH, Oberkochen, Germany, equipped with Plan-NeoFluar 10×/0.3 NA, Plan-NeoFluar 40×/0.75 NA, and Plan-Apochromat 63×/1.4 NA objectives) with ApoTome, as well as a Nikon C2 confocal microscope (with a 60× Oil Plan APO VC NA = 1.45 objective) (Tokyo, Japan). For data analysis and evaluation, AxioVision 4.82 and ImageJ 1.52c software were used.

### 4.7. Statistics

Statistical analyses were performed using GraphPad Prism 8.0.1. Significance between the experimental and control groups was determined using one-way ANOVA or *t*-test (Appendix A).

## 5. Conclusions

Our study highlights that autophagic activity in *C. elegans* is a highly dynamic process, responding rapidly to changes in both internal and external factors. This regulation ensures efficient recycling of cellular components during stress and supports survival under challenging environments. Dysregulation of autophagy is linked to numerous human diseases, including neurodegeneration, metabolic disorders, and cancer. The newly developed GFP::mCherry::LGG-1 and GFP::ATG-5 reporters provide reliable in vivo systems to monitor autophagic flux, and *C. elegans* serves as a powerful model to investigate the molecular mechanisms of autophagy and its relevance to disease.

## Figures and Tables

**Figure 1 ijms-26-10178-f001:**
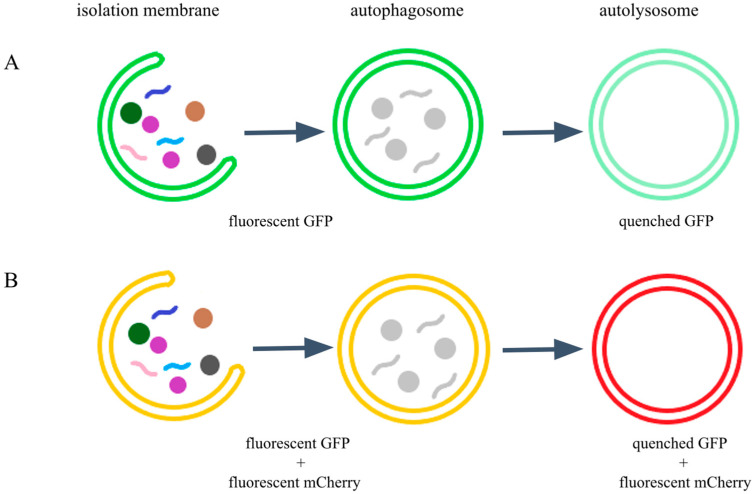
Function of GFP and mCherry labeling in autophagosomes and autolysosomes. (**A**) GFP labels autophagosomes in green (punctate green fluorescent signal); however, due to the acidic pH of the lysosome, GFP is quenched in the autolysosome, and thus the GFP signal disappears (no green fluorescence is visible in this structure after fusion). (**B**) The combined presence of green GFP and red fluorescent mCherry produces a yellow signal in autophagosomes (co-localization). Following fusion with the lysosome, GFP is quenched in the acidic lumen, leaving only the red mCherry fluorescence visible. In this way, our endogenous reporter distinguishes between different stages of autophagy. (Colored dots and lines represent autophagic substrates, while grey ones represent degrading substrates).

**Figure 2 ijms-26-10178-f002:**
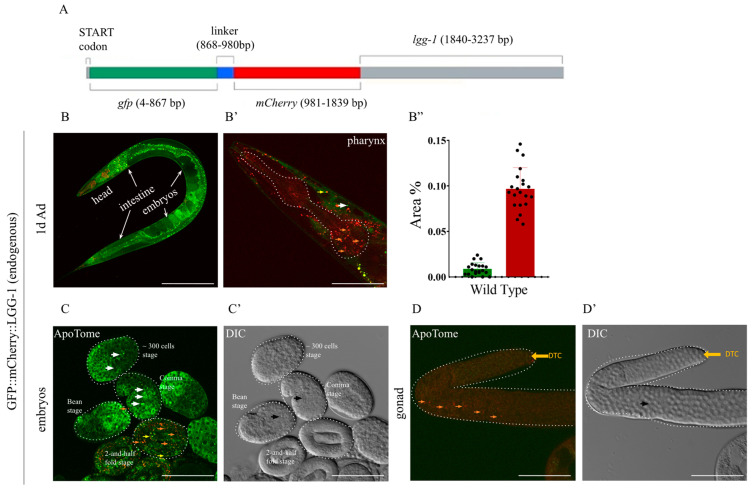
The endogenously tagged GFP::mCherry::LGG-1 reporter is expressed from early embryonic stages throughout adulthood. (**A**) The *gfp::mcherry::* linker coding sequence was inserted after the START (ATG) codon of the endogen *lgg-1* gene. (**B**) Fluorescent stage of the tandem GFP::mCherry::LGG-1 reporter. In merged images of animals expressing GFP::mCherry::LGG-1, autophagosomes (AP) are marked by yellow and autolysosomes (AL) are marked by red dots. The scale bars represent 200 μm. Fluorescent signals: GFP (green), mCherry (red). (**B’**) Representative picture of endogenously tagged GFP::mCherry in adult hermaphrodites and head with the pharynx (dashed line). Orange arrowhead AL, yellow arrowhead AP, white arrowhead GFP only dots. The scale bars represent 50 μm. Fluorescent signals: GFP (green), mCherry (red). (**B”**) Quantification of the percentage of the autophagy structures-covered area in the 1-day-old adult head. Data is presented as mean and SD of three independent measurements with min. 10 animals in each. Each black dot represents a measured individual. For statistics, see Appendix A. (**C**,**C’**) Representative picture of endogenously tagged GFP::mCherry at different embryonic stages with ApoTome (**C**) and DIC (**C’**) imaging. The scale bars represent 50 μm. Fluorescent signals: GFP (green), mCherry (red). Orange arrowhead AL, yellow arrowhead AP, white arrowhead GFP only dots, black arrowhead marks apoptotic corps. (**D**,**D’**) Representative picture of endogenously tagged GFP::mCherry in adult gonad with ApoTome (**D**) and DIC (**D’**) imaging. Distal tip cells (DTC) are marked by a yellow arrow. Orange arrowhead AL, black arrowhead points to apoptotic corps. The scale bars represent 50 μm. Fluorescent signals: GFP (green), mCherry (red).

**Figure 3 ijms-26-10178-f003:**
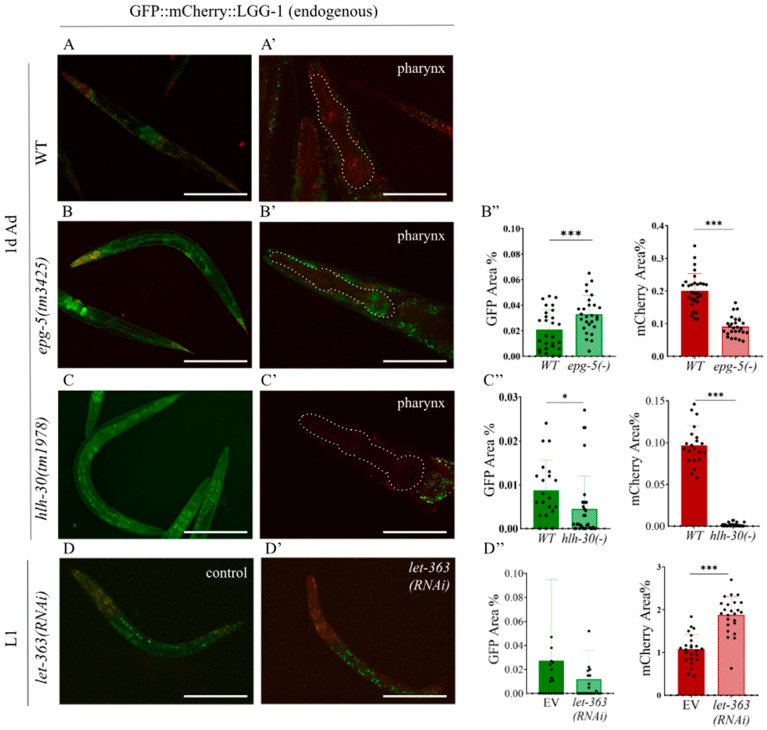
Expression of the endogenously tagged GFP::mCherry::LGG-1 reporter is highly responsive to genetic alterations that change autophagic activity. (**A**–**C**) Representative pictures of endogenously tagged GFP::mCherry::LGG-1 expression in adult hermaphrodites (**left**) and in the head (**right**) (the pharynx is outlined by a dashed line) in wild-type (**A**), *epg-5(-)* (**B**), and *hlh-30(-)* (**C**) mutant animals. The scale bars represent 200 μm in the whole-body image and 50 μm in the pharyngeal image. Fluorescent signals: GFP (green), mCherry (red). (**D**,**D’**) Representative picture of endogenously tagged GFP::mCherry::LGG-1 expression in an L1 larva treated with control (**D**) and *let-363* RNAi (**D’**), pharynx marked by a dashed line. The scale bars represent 50 µm. Fluorescent signals: GFP (green), mCherry (red). (**B”**–**D”**) Quantification of the autophagy pool size in the pharynx of the 1-day-old adult hermaphrodites in *epg-5(-)* (**B”**) and *hlh-30(-)* (**C”**) mutants *vs*. wild-type animals, and in the pharynx of the control and *let-363* RNAi-treated L1 animals (**D”**). GFP represents the area of AP pool only, mCherry represents the total pool of autophagy structures (AP + AL). Data are the mean ± SD of > X animals, data are combined from three independent experiments. Statistical significance was determined using non-paired *t*-test, * *p* < 0.01, *** *p* < 0.0001. For statistics, see Appendix A.

**Figure 4 ijms-26-10178-f004:**
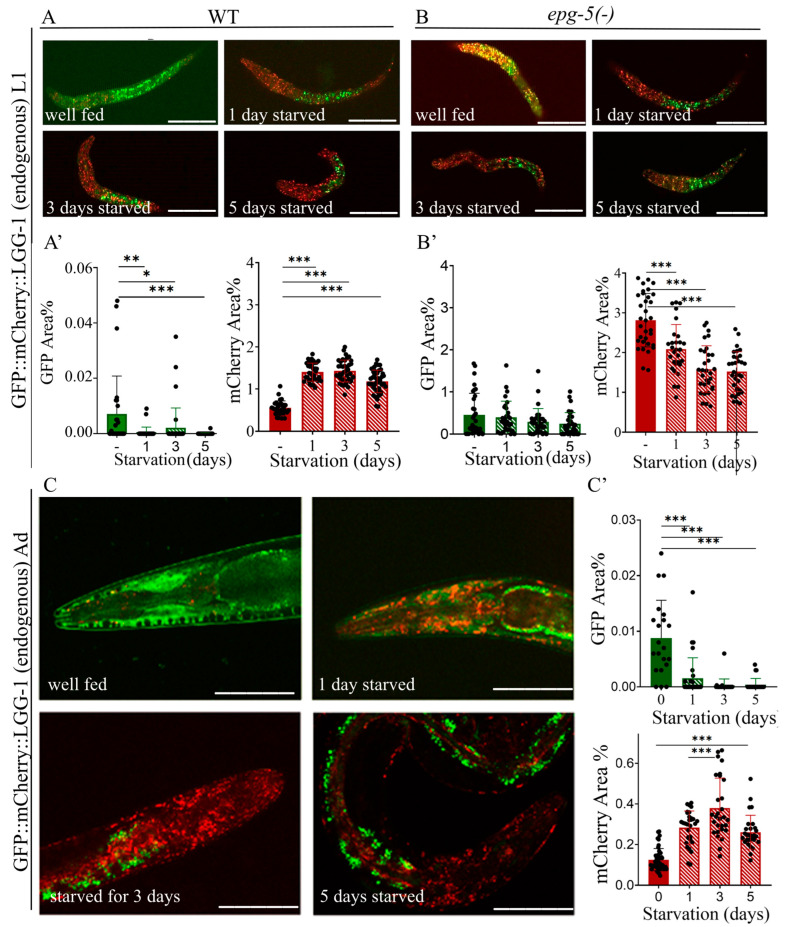
The quantity of mCherry-labeled autophagy structures increases in response to starvation at the L1 larval stage and young adulthood in an otherwise wild-type genetic background, but decreases in autophagy-defective *epg-5(-)* mutant genetic background. (**A**,**B**) Representative images of GFP::mCherry::LGG-1 transgene expression in wild type (**A**) and *epg-5(tm3425)* mutant (**B**) L1 larvae showing the relative abundance of GFP- and mCherry-labeled structures. The scale bars represent 50 μm. Fluorescent signals: GFP (green), mCherry (red). (**A’**,**B’**) Quantification of the percentage coverage of GFP- and mCherry-labeled structures in the head region of GFP::mCherry::LGG-1 transgene expression in wild type (**A’**) and *epg-5(tm3425)* mutant (**B’**) L1 larvae. GFP represents the autophagosome (AP) pool only, while mCherry represents the total autophagic structures (AP + AL). Each black dot represents a measured individual. Statistical significance was determined using an unpaired *t*-test, * *p* < 0.01, ** *p* < 0.001, *** *p* < 0.0001. (**C**) Representative images of GFP- and mCherry-labeled structures in young adult animals expressing GFP::mCherry::LGG-1 under fed and starved conditions. The head region of the animals is shown. The scale bars represent 50 μm. Fluorescent signals: GFP (green), mCherry (red). (**C’**) Quantification of the percentage coverage of GFP- and mCherry-labeled structures in the head region of GFP::mCherry::LGG-1 young adults. GFP represents the AP pool only, while mCherry represents the total pool of autophagic structures (AP + AL). Each black dot represents a measured individual. Statistical significance was determined using an unpaired *t*-test, *** *p* < 0.0001. For statistics, see Appendix A.

**Figure 5 ijms-26-10178-f005:**
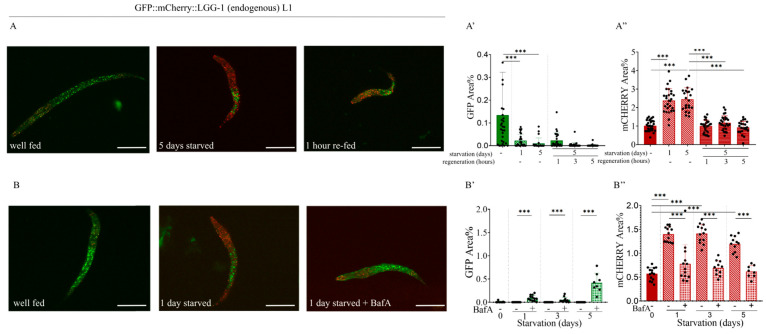
In response to bafilomycin A1 treatment, the quantity of mCherry-labeled autophagy structures in starved L1 larvae decreases to levels that are similar to those found in well-fed animals. (**A**) Representative images of GFP::mCherry::LGG-1 L1 larvae showing GFP- and mCherry-labeled structures under fed conditions, after 5 days of starvation, and after refeeding following 5 days of starvation. The scale bars represent 50 μm. Fluorescent signals: GFP (green), mCherry (red). (**B**) Representative images of GFP::mCherry::LGG-1 L1 larvae showing GFP- and mCherry-labeled structures under fed conditions, after 1 day of starvation, and after 1 day of starvation with bafilomycin A1 treatment. The scale bars represent 50 μm. Fluorescent signals: GFP (green), mCherry (red). (**A’**,**A”**,**B’**,**B”**) Quantification of the percentage area coverage of GFP- (**A’**,**B’**) and mCherry-labeled (**A”**,**B”**) structures in the head region of GFP::mCherry::LGG-1 L1 larvae is shown (graphs, right panels). GFP represents the autophagosome (AP) pool only, while mCherry represents the total pool of autophagic structures (AP + AL). Each black dot represents a measured individual. Statistical significance was determined using an unpaired *t*-test, *** *p* < 0.0001. For statistics, see Appendix A.

**Figure 6 ijms-26-10178-f006:**
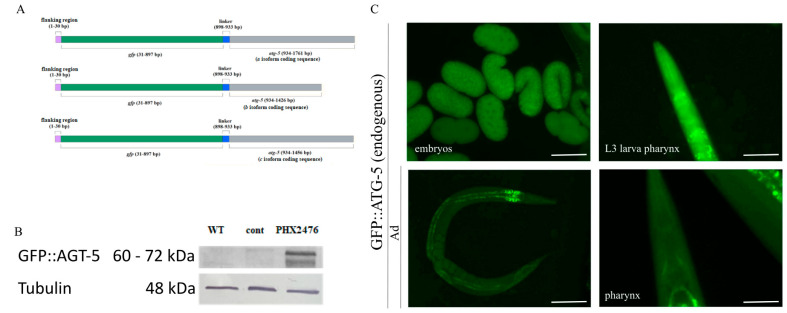
Expression analysis of the endogenous GFP::ATG-5 reporter. (**A**) Schematic of the GFP::ATG-5 reporter construct. The GFP sequence is positioned upstream of the *atg-5* coding sequence in all three isoforms. (**B**) Western blot analysis detecting the GFP::ATG-5 protein. The protein was detected in PHX2476 animals but not in wild-type animals or in those expressing another endogenous GFP gene used as a control. Tubulin was used as a loading control. WT: wild type; EV: empty vector; cont (control): endogenous GFP-expressing strain; kDa: kilo-Dalton. (**C**) GFP::ATG-5 reporter accumulation at different embryonic stages (**top left**), in the pharynx of an L3 stage larva (**top right),** in a young hermaphrodite (**bottom left**), and in the pharynx of an adult hermaphrodite (**bottom right**). Exposure time: 200 ms. The scale bars represent 50 μm in the embryo image and in the pharyngeal images, and 20 μm in the whole-body image. Fluorescent signal: GFP (green).

**Figure 7 ijms-26-10178-f007:**
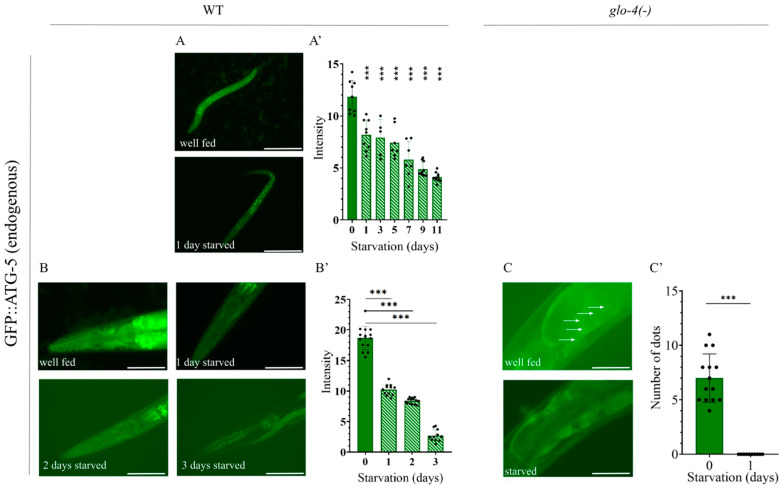
The endogenous GFP::ATG-5 reporter is responsive to starvation-induced stress. (**A**) Representative images of GFP::ATG-5 L1 larvae under fed and 1-day starved conditions. Exposure time: 800 ms. The scale bars represent 50 μm. Fluorescent signal: GFP (green). (**A’**) Quantification of GFP expression intensity in the head region of GFP::ATG-5 L1 larvae. GFP represents the autophagosome (AP) pool. Each black dot represents a measured individual. Statistical significance was determined using an unpaired *t*-test, *** *p* < 0.0001. (**B**) Representative images of young adult animals expressing the endogenous GFP::ATG-5 reporter under fed conditions and after 1, 2, and 3 days of starvation. Images show the head region of the animals. Exposure time: 800 ms. The scale bars represent 50 μm. Fluorescent signal: GFP (green). (**B’**) Quantification of GFP expression intensity in the head region of GFP::ATG-5 young adults. GFP represents the AP pool. Each black dot represents a measured individual. Statistical significance was determined using an unpaired *t*-test, *** *p* < 0.0001. (**C**) Representative images of the anterior intestine of young adult GFP::ATG-5 animals under fed and starved conditions. Arrows indicate GFP-positive autophagic structures in the first intestinal cell of fed animals. Exposure time: 1500 ms. The scale bars represent 10 μm. Fluorescent signal: GFP (green). (**C’**) Quantification of GFP-labeled structures in the anterior intestine of GFP::ATG-5 young adults in the *glo-4(-)* mutant genetic background. Each black dot represents a measured individual. Statistical significance was determined using an unpaired *t*-test, *** *p* < 0.0001. For statistics, see Appendix A.

## Data Availability

The original contributions presented in this study are included in the article/Appendix A. further inquiries can be directed to the corresponding author(s).

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
