# Peer review of "Developing Endogenous Autophagy Reporters in Caenorhabditis elegans to Monitor Basal and Starvation-Induced Autophagy"

_ijms, 2025, doi:10.3390/ijms262010178_

Round 1
Reviewer 1 Report
Comments and Suggestions for Authors
In this study, the authors developed endogenous autophagy reporters in C. elegans and monitored both basal and starvation-induced autophagy using this newly established detection system. CRISPR-Cas9–mediated knock-in of tag proteins such as GFP, RFP, or Flag has recently been widely applied to investigate endogenous gene regulation. Similarly, in the autophagy field, knock-in studies such as GFP-LC3 in 293 cells (2018, Biotechnol J.) and tandem-fluorescent LC3 (tfLC3) mice (2025, autophagy) have been successfully established to monitor autophagy activity. In this manuscript, the authors generated C. elegans strains harboring GFP-mCherry-LC3 and GFP-Atg5 endogenous reporters. These tools are expected to become a valuable resource for studying autophagy in C. elegans, and this work represents a timely and important contribution to the field.
Minor concerns
- The authors are encouraged to show the mRNA expression levels of GFP, mRFP, and LC3 under both basal and starvation conditions using qPCR. In addition, western blotting should be performed to compare the protein levels of endogenous LC3 with mCherry-GFP-tagged LC3, which would help confirm that the tagged construct faithfully reflects endogenous LC3 regulation.
- In Figure 7C, the authors should carefully verify the size of GFP-Atg5. Although Atg5 itself is approximately 30 kDa, it is typically detected around 55 kDa due to covalent conjugation with Atg12. Please check whether GFP-Atg5 is similarly conjugated with Atg12 and show appropriate controls or immunoblots to confirm this. This verification is essential to ensure that the reporter accurately reflects physiological Atg5 function.
Author Response
The authors are encouraged to show the mRNA expression levels of GFP, mRFP, and LC3 under both basal and starvation conditions using qPCR. In addition, western blotting should be performed to compare the protein levels of endogenous LC3 with mCherry-GFP-tagged LC3, which would help confirm that the tagged construct faithfully reflects endogenous LC3 regulation.
Answer: The endogenous gfp::mCherry::lgg-1 transgenic strain was created by inserting the two reporter genes (gfp and mCherry) into the endogenous lgg-1 gene. Thus, the transgenic strain also contains a single (endogenous) lgg-1 gene, but not an extra copy of the gene; wild-type and transgenic strains are identical in that both of them have a single (endogenous) lgg-1 sequence.
Because we have not LGG-1/LC3-specific antibody, we were unable to perform the WB analysis that was suggested by Ref 1. However, we performed a qPCR analysis to compare the endogenous lgg-1 expression levels between the two strains, wild-type (control) and transgenic ones. The qPCR analysis (Figure 1, here) showed that the transgenic strain has a slightly lower level of lgg-1 expression, around 50% (0.46 ratio), as compared to wild-type when animals were propagated under normal, well-fed conditions. A similar difference (0.58 ratio) in lgg-1 expression levels could also be observed between the two strains under starvation-induced conditions. This difference could be the result of reporter gene insertion into the upstream region of the endogenous lgg-1 sequence, just after the translational START site, which may interfere with the effects of transcription factors.
Figure 1. lgg-1 expression in the transgenic strain (PHX2459) is lower by 50% as compared to that found in the wild type (N2). The figure shows the ratio (0.46 – the dark grey bar) of lgg-1 expression levels in wild-type vs. transgenic animals maintained under normal (fed) conditions. When animals were maintained under starved conditions, a similar ratio (0.58 – the light grey bar) could be observed.
Corresponding data:
|
lgg-1 expression levels in PHX2459 (transgenic) vs. N2 (control) |
ddC |
Relative Quantity |
|
fed |
1.11 |
0.4632940309 |
|
starved |
0.78 |
0.5823667932 |
qPCR method
qPCR
Gravid adult wild-type (N2) or PHX2459 lgg-1(syb2459) animals were allowed to lay eggs for 4-8 hours to obtain a synchronized population on NGM plates seeded with OP50 and maintained at 20°C until the animals reached the L4 stage. Starved animals were incubated in M9 for 24 hours at 20°C, while control animals were left on OP50 plates. After incubation, animals were collected and washed with M9 buffer.
RNA was isolated using RNAzol® RT (RN 190) (Molecular Research Center, INC.; 5645 Montgomery Road, Cincinnati, OH 45212, USA) and purified from the aqueous phase after extraction using RNA Clean & ConcentratorTM-5 kit (R1013) (Zymo Research Co.; 17062 Murphy Ave., Irvine, CA 92614, USA). cDNA was synthesized using the RevertAid First Strand cDNA Synthesis Kit (K1622) (Thermo Fisher Scientific Inc.; 81 Wyman St, Waltham, MA 02451, USA).
Quantitative real-time PCR was performed using the Roche LightCycler® 96 System (F. Hoffmann-La Roche AG, Grenzacherstrasse 124, 4070 Basel, Switzerland) using the following primers: lgg-1 Forward 5’-ACC CAG ACC GTA TTC CAG TG-3’, Reverse 5’-ACG AAG TTG GAT GCG TTT TC-3’, cdc-42 (internal control) Forward 5’-CTG CTG GAC AGG AAG ATT ACG-3’, Reverse 5’-CTC GGA CAT TCT CGA ATG AAG-3’, gfp Forward 5’-TCT GTC AGT GGA GAG GGT GAA-3’, Reverse 5’-GAC AAG TGT TGG CCA TGG AAC-3’.
Cycle numbers and relative levels (adjusted to cdc-42), one biological sample, 3 technical parallels with negative and positive controls).
|
cdc-42 |
lgg-1 |
gfp |
dC lgg-1 |
dC gfp |
|
|
PHX2459 fed |
31,19 |
30,18 |
30,5 |
-1,01 |
-0,69 |
|
PHX2459 starved |
22,33 |
20,62 |
20,51 |
-1,71 |
-1,82 |
|
N2 fed |
23,82 |
21,7 |
39,76 |
-2,12 |
15,94 |
|
N2 sarved |
30,16 |
27,67 |
36,29 |
-2,49 |
6,13 |
The ratio of gfp:lgg-1 expression levels. The ratio tends to 1 (ideal).
Figure 2. The ratio of gfp:lgg-1 expression levels. Animals were maintained under well-fed and starved conditions. Wild-type animals do not contain gfp transgene, so the level everywhere is zero.
|
lgg-1/gfp expression levels. |
ddC |
RQ |
|
PHX2459 (transgenic) fed |
0.32 |
0.801 |
|
PHX2459 starved |
-0.11 |
1.079 |
|
N2 (wild-type) fed |
18.06 |
0.00000365 |
|
N2 sarved |
8,62 |
0.0025 |
Below, we compared the effects of starvation and well-feeding on lgg-1 expression levels in transgenic (PHX2459) and wild-type (N2) animals.
Figure 3. RQ (relative quantity) data of lgg-1 expression in starved vs. well-fed animals.
|
ddC |
RQ |
|
|
PHX2459 |
-0.7 |
1.6245 |
|
N2 |
-0.37 |
1.2923 |
After a 24 hour-long starvation period, lgg-1 expression increased by 1.5 times in both transgenic (PHX2459) and control (N2) strains as compared to well-fed conditions. These data are similar to those reported previously by Harvald et al. (PMID: 28734827).
The original WB analysis we performed showed that GFP::mCherry-LGG-1 protein accumulation decreases to a minimal level
Figure 3. LGG-1 protein levels in well-fed and starved animals. WB analysis shows that LGG-1 levels are detectable in the non-silenced animals maintained under normal conditions (the left red arrow), while become very low in non-silenced animals exposed to starvation (the right red arrow). EV: empty vector (non-silenced).
In Figure 7C, the authors should carefully verify the size of GFP-Atg5. Although Atg5 itself is approximately 30 kDa, it is typically detected around 55 kDa due to covalent conjugation with Atg12. Please check whether GFP-Atg5 is similarly conjugated with Atg12 and show appropriate controls or immunoblots to confirm this. This verification is essential to ensure that the reporter accurately reflects physiological Atg5 function.
Answer: Based on this suggestion, we have reevaluated the corresponding WB image.
MW: Precision Plus Protein Unstained Standards
1: N2 anti-GFP
2: endogenous GFP tagged control with anti-GFP
3: endogenous GFP::ATG-5 with anti-GFP
4: N2 anti-Tubulin (49 kDa)
5: endogenous GFP tagged control anti-Tubulin (49 kDa)
6: endogenous GFP::ATG-5 anti-Tubulin (49 kDa)
Figure 3. It is visible that between 60-72 kDa there are 3 different bands, which correspond to the 3 GFP-tagged ATG-5 isoforms conjugated to ATG-12 and LGG-3. These transgenic animals showed a superficially wild-type phenotype.
Reviewer 2 Report
Comments and Suggestions for Authors
- Reduce introduction part in abstarct and add more results in it.
- There is confusion between the strain names provided by Suny Biotech and the lab's internal names.
- For each graph, state the exact n number (e.g., n=15, 20, etc.) for each condition/group, specifying if it represents biological (number of worms) or technical replicates. Clearly state which specific statistical test was used (e.g., unpaired t-test, one-way ANOVA with Tukey's post-hoc test) in the figure legend or methods.
- Is there any need of section 2.2. Sequences of the endogenous autophagy reporters? It can be shifteed in supplimentary data.
- The concentration of Bafilomycin A1 used is not stated. This is crucial for reproducibility.
- The duration of bleach treatment to dissolve adults should be specified (e.g., "incubated for 5-7 minutes with occasional vortexing").
- The description of the F1 and F2 generations is slightly confusing. Clarify that the F1 generation was exposed to RNAi from hatching, and the F2 generation was both conceived and raised entirely on RNAi bacteria, leading to a stronger (possibly maternal and zygotic) knockdown.
- Conclusion part is absent from manuscript.
Author Response
- Reduce introduction partin abstarct and add more results in it.
Answer: The manuscript has been changed accordingly.
- There is confusion between the strain names provided by Suny Biotech and the lab's internal names.
Answer: We have left the name of the original transgenic strain generated by Suny Biotech Ltd., PHX2459. However, when we generated double mutant strains with this transgenic strain, we followed the strain nomenclature of our lab, TTV. For example, the strain name TTV823 refers to atg-5(syb2476) [atg-5p::gfp::atg-5]; glo-4(ok623)V genotype.
- For each graph, state the exact n number (e.g., n=15, 20, etc.) for each condition/group, specifying if it represents biological (number of worms) or technical replicates. Clearly state which specific statistical test was used (e.g., unpaired t-test, one-way ANOVA with Tukey's post-hoc test) in the figure legend or methods.
Answer: All statistical data are now available in the Supplementary Materials file that was not uploaded during the original submission.
- Is there any need of section 2.2. Sequences of the endogenous autophagy reporters? It can be shifteed in supplimentary data.
Answer: The manuscript has been changed accordingly.
- The concentration of Bafilomycin A1 used is not stated. This is crucial for reproducibility.
Answer: The final concentration of Baf A1 was 30 nM, and this information has been added to the Material and Methods section.
- The duration of bleach treatment to dissolve adults should be specified (e.g., "incubated for 5-7 minutes with occasional vortexing").
Answer: Animals were incubated for 8-10 minutes with accessional vortexing. When adults became invisible, 5 ml of M9 solution was added to stop the bleaching reaction. This information has been added the Material and Methods section.
- The description of the F1 and F2 generations is slightly confusing. Clarify that the F1 generation was exposed to RNAi from hatching, and the F2 generation was both conceived and raised entirely on RNAi bacteria, leading to a stronger (possibly maternal and zygotic) knockdown.
Answer: The manuscript has been supplemented by the following information: RNA interference has no phenotypic effect on the F1 generation, which had been exposed to RNAi bacteria from hatching.
- Conclusion part is absent from manuscript.
Answer: The following Conclusion has been added to the manuscript.
- Conclusion
Our study highlights that autophagic activity in C. elegans is a highly dynamic process, responding rapidly to changes in both internal and external factors. This regulation ensures efficient recycling of cellular components during stress and supports survival under challenging environments. Dysregulation of autophagy is linked to numerous human diseases, including neurodegeneration, metabolic disorders, and cancer. The newly developed GFP::mCherry::LGG-1 and GFP::ATG-5 reporters provide reliable in vivo systems to monitor autophagic flux, and C. elegans serves as a powerful model to investigate the molecular mechanisms of autophagy and its relevance to disease.
Round 2
Reviewer 2 Report
Comments and Suggestions for Authors
Please add research gap and novelty in abstract.
Author Response
Borden et al. 2025
Response to Reviewer 2
Thank you very much for taking the time to review this manuscript. Please find the corrections highlighted in the re-submitted files.
Comments and Suggestions for Authors - Please add research gap and novelty in abstract.
Revised abstract
Autophagy (cellular self-eating) is a tightly regulated catabolic process of eukaryotic cells during which parts of the cytoplasm are sequestered and subsequently delivered into lysosomes for degradation by acidic hydrolases. This process is central to maintaining cellular homeostasis, the removal of aged or damaged organelles, and the elimination of intracellular pathogens. The nematode Caenorhabditis elegans has proven to be a powerful genetic model for investigating the regulation and mechanism of autophagy. To date, the fluorescent autophagy reporters developed in this organism have predominantly relied on multi-copy, randomly integrated transgenes. As a result, the interpretation of autophagy dynamics in these models has required considerable caution due to possible overexpression artifacts and positional effects. In addition, starvation-induced autophagy has not been characterized in details using these reporters. Here, we describe the development of two endogenous autophagy reporters, gfp::mCherry::lgg-1/atg-8 and gfp::atg-5, both inserted precisely into their endogenous genomic loci. We demonstrate that these single-copy reporters reliably track distinct stages of the autophagic process. Using these tools, we reveal that (i) the transition from the earliest phagophore to the mature autolysosome is an exceptionally rapid event because the vast majority of the detected fluorescent signals are autolysosome-specific, (ii) starvation triggers autophagy only after a measurable lag phase rather than immediately, and (iii) the regulation of starvation-induced autophagy depends on the actual life stage, and prevents excessive flux that could otherwise compromise cellular survival. We anticipate that these newly developed reporter strains will provide refined opportunities to further dissect the physiological and pathological roles of autophagy in vivo.
